# Co-Occurrence of a Pathogenic *HSD3B2* Variant and a Duplication on 10q22.3-q23.2 Detected in Newborn Twins with Salt-Wasting Congenital Adrenal Hyperplasia

**DOI:** 10.3390/genes13122190

**Published:** 2022-11-23

**Authors:** Simona Mellone, Enrica Bertelli, Barbara Roviglione, Denise Vurchio, Sara Ronzani, Andrea Secco, Enrico Felici, Mariachiara Martina Strozzi, Federico Schena, Mara Giordano

**Affiliations:** 1Laboratory of Genetics, Clinical Biochemistry Unit, University Hospital Maggiore della Carità, 28100 Novara, Italy; 2Pediatric and Pediatric Emergency Unit, Children Hospital, Azienda Ospedaliera SS Antonio e Biagio e C. Arrigo, 15121 Alessandria, Italy; 3Department of Health Sciences, Università del Piemonte Orientale, 28100 Novara, Italy; 4Neonatal Intensive Care Unit, Azienda Ospedaliera SS Antonio e Biagio e C. Arrigo, 15121 Alessandria, Italy

**Keywords:** *HSD3B2*, congenital adrenal hyperplasia, aCGH-SNP, LCSH

## Abstract

Congenital adrenal hyperplasia (CAH) is a group of autosomal recessive disorders caused by enzyme deficiencies required for cortisol biosynthesis in the adrenal cortex. The majority of CAH are due to the deficiency of the 21-hydroxylase enzyme, while 3β-hydroxysteroid dehydrogenase type 2 deficiency accounts for less than five percent of all CAH cases. We report two Moroccan twins from a spontaneous triplet pregnancy. The 46,XY newborn exhibited a disorder of sexual differentiation (DSD) with hypo virilization, while the 46,XX newborn had normal female external genitalia. In the first week of life, they showed hyponatremia and primary adrenal insufficiency with a slight 17OHP elevation and increased DHEAS and renin levels. The aCGH-SNP analysis disclosed a 8.36 Mb long contiguous stretch of homozygosity (LCSH) on chromosome 1p13.2-p11.2 including the candidate *HSD3B2* gene, a LCSH of 7.3 Mb on 14q31.1-q32.11, and a 7 Mb duplication on 10q22.3-q23.2. Clinical exome sequencing revealed the biallelic c.969T > G (p.Asn323Lys) *HSD3B2,* likely pathogenic, variant in both of the affected twins. This case emphasizes the importance of a prompt molecular diagnosis performed through the combination of aCGH and clinical exome, both for establishment of correct therapy and for follow-up, as the newborns also carry a genomic rearrangement with possible clinical implications.

## 1. Introduction

Congenital adrenal hyperplasia (CAH) refers to a group of autosomal recessive disorders caused by single gene defects in one of the enzymes required for cortisol biosynthesis [1]. The most frequent genetic defect consists in a 21-hydroxylase deficiency (21-OHD), responsible for more than 95% of all cases of CAH, followed by 11-hydroxylase deficiency (11-OHD) that accounts for 0.2–8% of cases [2]. A rare form of CAH is represented by 3β-hydroxysteroid dehydrogenase type 2 deficiency (*HSD3B2*; MIM 201810) that accounts for less than 0.5% of all CAH cases, with an estimated prevalence at birth of <1/1,000,000 [3]. This disorder is caused by biallelic HSD3B2 gene alterations that impair steroid biosynthesis in both the adrenal glands and gonads. In humans, there are two isoforms of 3β-hydroxysteroid dehydrogenase, encoded by two different genes, located on 1p13.1 that share 93.5% homology. The HSD3B1 gene (type 1 isoenzyme 3βHSD1), is expressed in the placenta and peripheral tissues, and the isozyme catalyzes transformation of dehydroepiandrosterone (DHEA) into sex steroids. *HSD3B2* encodes the adrenal and gonadal 3βHSD2 enzyme, essential for the biosynthesis of steroid hormones, including aldosterone, cortisol, and sex steroids [4]. It is responsible for the conversion of the Δ5-steroids (pregnenolone, 17-hydroxypregnenolone, and DHEA) to Δ4-steroids (progesterone, 17-hydroxyprogesterone (17-OHP), and androstenedione), respectively. Consequently, in patients with loss of HSD3B2 activity, high ratios of the Δ5-steroids over Δ4-steroids are the main hormonal changes observed, with circulating DHEA being converted to testosterone by the intact extra-adrenal 3βHSD1 [5]. DHEA is also converted by SULT2A1 to the more stable sulfated form (DHEAS) that is measured more often than DHEA for practical reasons [3]. HSD3B2 gene mutations cause steroid hormone synthesis disorder that eventually leads to decreased aldosterone, cortisol, and sex hormone synthesis. The severity of the phenotype depends mainly on the remaining enzyme activity including salt loss in neonates, incomplete masculinization of males, mild virilization or normal external genitalia of females, premature pubarche and/or menstrual disorders, and/or various degrees of hypogonadism in adolescents. Genital abnormalities, small penis, and hypospadias in 3βHSD2D are caused by impaired testosterone biosynthesis due to the lack of 3βHSD2 in the testes, during the critical period of external genital development before the 12th week of gestation during the fetal period [6]. The phenotype also depends on the type of genetic defect; in general, frameshift, splicing, and nonsense mutations are associated with severe forms of HSD3B2 deficiency resulting in salt-wasting (SW) crisis phenotype manifested through vomiting and dehydration, while missense variants are the most common type, associated with some residual enzymatic activity and non-SW form [7,8] as about 10% of residual activity is sufficient to prevent aldosterone deficiency [9].

Here we describe the clinical data, hormonal pattern, and the results of the genetic analysis of two newborn Moroccan twins from a spontaneous triplet pregnancy, a male and a female, respectively, presenting with a disorder of sex development (DSD) and normal female genitalia. Hormonal findings in both revealed a SW-CAH, however, the clinical pattern was not typical for 21-OHD, suggesting the need to investigate a less common form of CAH. In SW crisis, emergency treatment includes prompt rehydration, correction of hypoglycemia, and parenteral hydrocortisone. The maintenance treatment for 3βHSD2D is hormone replacement in patients who fail to enter or progress through puberty [3], and adequate NaCl intake. Surgical corrective procedures may be indicated in case of undescended testes, hypospadias repair in males, and severe genital virilization in females. The third newborn had normal male external genitalia and remained asymptomatic. The affected siblings carried a biallelic, likely pathogenic, variant in *HSD3B2* in addition to a 7 Mb duplication on 10q22.3-q23.2 revealed by array CGH.

## 2. Materials and Methods

### 2.1. Case Presentation

Three newborns (2 males, 1 female) were admitted to NICU for prematurity and low birth weight at 33 + 1 weeks of gestational age from cesarean section after spontaneous triplet pregnancy. Their parents, native from a small county in Morocco, did not refer consanguinity and were completely asymptomatic as well as the two older brothers, respectively 11 and 8 years old. The first newborn (the male) showed a birth weight 1930 g, APGAR 9/10, spontaneous breathing, and jaundice treated with phototherapy. He manifested a disorder of sexual differentiation (DSD): Prader IV with perineal hypospadias, small penis, and hyper pigmented and fused scrotal folds containing palpable oval formations (Figure 1). His karyotype was 46,XY, suggesting a 46,XY DSD with hypovirilization of external genitalia. At birth the female weighed 1950 g with APGAR 7/8, mild respiratory distress, and jaundice requiring phototherapy; she presented normal female external genitalia and 46,XX karyotype. Both the twins maintained normal blood pressure and normoglycemia, but reported hyponatremia in the first week of life (respectively 133 and 129 mEq/L), poor weight gain, and progressive eating difficulties (regurgitation and vomiting requiring, in the male, enteral feeding by nasogastric tube for 15 days). The third newborn (a male, birth weight 2130 g, APGAR 8/9) had normal male external genitalia and remained asymptomatic.

An abdomen US scan of the affected male revealed the absence of female internal genitalia and two oval hyper intense structures corresponding to the external genitalia, while the female had a uterus and ovaries; neither the patients showed any relevant ultrasonographic findings at the adrenal level.

Hormonal investigation in both the patients revealed primary adrenal insufficiency with a slight 17-OHP elevation and a marked increase of DHEAS and renin (Table 1). Adrenal function was normal in all the other siblings as well as pituitary, thyroid, and gonadal functions (data not shown).

The two affected newborns started IV hydrocortisone at 50 mg/m^2^/day (gradually decreased to an oral maintenance dose of 10–15 mg/m^2^/day), fludrocortisone 0.1 mg/day, and NaCl 2–5 mEq/kg/die orally administered, allowing regular growth and satisfactory control of adrenal insufficiency without hypertension. Two-stage surgical correction of hypospadias was scheduled at age 12 and 18 months for the male.

### 2.2. CGH-SNP Microarray Analysis

Following written informed consent, genomic DNA of all the family members was extracted from peripheral blood through the ReliaPrep Blood gDNA Miniprep System (Promega), according to the manufacturer’s recommendation.

Medium-density microarray analysis, GenetiSure Dx Postnatal Array 4 × 180 K + SNP (Agilent Technologies, Santa Clara, CA, USA) was performed using standard protocols. The SNP array contains ~59,000 single-nucleotide polymorphism probes and ~107,000 oligonucleotide probes (60-mer) with a resolution of 5~10 Mb for ROH detection.

The slide was scanned through Agilent SureScan Dx Microarray Scanner System (Agilent Technologies, Santa Clara, CA, USA).

Image analysis, normalization, and annotation were based on Agilent Feature Extraction Software. Finally, CNVs were called with Agilent CytoDx Software 1.1.1.0 using tiff images from data.

### 2.3. Clinical Exome Sequencing (CES)

Library preparation was performed using the SureSelectQXT NGS target enrichment kit for Illumina multiplexed sequencing and the Agilent SureSelect Custom Constitutional Panel 17 Mb (Agilent, Santa Clara, CA, USA), according to manufacturer instructions. Sequencing probes cover all coding exons ±20 bp flanking sequence from the intron-exon boundary of 5,227 clinically relevant genes. Exon-enriched library was subjected to a 150 bp paired-end sequencing on the Illumina MiSeq platform (Illumina, San Diego, California, CA, USA). Sequencing reads passing quality filters were aligned to the human reference genome build (GRGh37/hg19), and variant calling was performed using the SureCall v3.5 software (Agilent Technologies). Then VCF files were annotated with the wANNOVAR tool. Finally, data was analyzed using a personalized bioinformatics pipeline. We considered only variants with a MAF (Minor Allele Frequency) <1% in public databases (1000 Genome project, ExAC and GnomAD) and all synonymous variants were excluded. Among the selected variants with this filtering, we have prioritized those that caused frameshift, stop codon, splicing, or amino acidic changes predicted as pathogenic by at least four in silico predictive tools (SIFT, Polyphen-2, Mutation Taster, CADD).

Pathogenicity of variants was determined following the consensus guidelines of the American College of Medical Genetics and Genomics, the Association for Molecular Pathology [10], and enGenome eVai software (evai.engenome.com) [11].

### 2.4. Sanger Sequencing

The *HSD3B2* variant identified by clinical-exome sequencing was validated by Sanger sequencing using the following primers designed by the Primer3 (v.0.4.0) software: forward: 5′-GCCTTCCTTTAACCCTGATGT-3′ and reverse: 5′-ACTCCACGGTTTTCTGCTTG-3′. The amplicon was purified using ExoSAP-IT™ Express PCR Product Cleanup Reagent (Applied Biosystems, Foster City, CA, USA), and then sequenced in the forward and reverse direction with the BigDye Terminator v1.1 Cycle Sequencing Kit (Applied Biosystems, Foster City, CA, USA) and analyzed on a SeqStudio Genetic Analyzer (Applied Biosystems).

### 2.5. MLPA (Multiplex Ligation-Dependent Amplification) Probe

To confirm the presence of the identified duplication, MLPA analysis was performed using the kit SALSA MLPA Probemix P200-B1 Reference-1 (MRC-Holland, Amsterdam, The Netherlands) according to manufacturer’s instructions. This includes ten reference probes that detect different autosomal chromosomal locations and four quality control fragments specific for unique human DNA sequences. At these probes we added a specific one designed in exon 1 of gene NRG3 (NM_001010848.3; start-end:83635049-83635849, assembly February 2009 GRCh37/hg19). The amplified fragments were analyzed by capillary electrophoresis on an Applied Biosystems SeqStudio Genetic Analyzer with the GeneMapper software (Applied Biosystems). The data analysis was performed by MRC-Holland Coffalyser v9.4 software.

## 3. Results

In order to investigate the presence of a molecular defect responsible for the salt-wasting and clinical presentations of the affected twins, the DNA of the proband was analyzed through array-CGH as a first-tier test.

The aCGH revealed a male karyotype with an interstitial 7 Mb duplication on chromosome 10q22.3-q23.2 (Figure 2). This region contains 17 genes OMIM, including *BMPR1A*, *NRG3*, *CDHR1* and *LDB3*. None of the genes included in the duplication were clearly implicated in the phenotype of the patient, although the precise function of all the involved genes is not yet completely clarified. The segregation analysis revealed that the duplication was inherited from the mother, transmitted to the affected sister (II-4, Figure 3), and to the healthy 8-year-old brother (II-2). Thus, this CNV was considered as a variant of uncertain significance (VUS).

The aCGH-SNP disclosed two long, continuous stretch of homozygosity (LCSH), one of 8.36 Mb (from rs2488762 to rs1853728) on chromosome 1p13.2-p11.2 containing 58 OMIM genes and the other of 7.27 Mb (from rs12435615 to rs11845372) on chromosome 14q31.1-q32.11 containing 12 OMIM genes (Figure 2). Among the genes included in these two LCSHs, the best candidate to explain the clinical characteristics of II-3 was *HSD3B2*, whose biallelic mutations have been previously detected in patients with 3-β-hydroxysteroid dehydrogenase 2 deficiency. In order to identify pathogenic variants in this gene, and eventually in other genes related to the disorder, a clinical-exome sequencing (CES) was performed.

The average sequencing depth was 64x with 97.43% of the target sequence covered at ≥20x, 91.74% covered at ≥50x and 73.61% covered at ≥100x. The variants were filtered as described and according to the ACMG guidelines. The two LCSH regions were confirmed by the CES, and a pathogenic homozygous variant on 1p13.2-p11.2 was detected in the exon 4 of *HSD3B2*, namely c.969T > G (Figure 3B). This missense variant converts codon 323 (AAT), which codes for Asparagine at highly conserved position, into AAG, encoding Lysine (p.Asn323Lys). The variant was not present in public databases (ExAC, dbSNP, ESP, gnomAD and the 1000 Genomes Project) and was classified as pathogenic according to the American College of Medical Genetics and Genomics (ACMG) guidelines for the interpretation of variations (criteria: PM2, PP1 Very Strong). No other clinically relevant variation was detected in other genes compatible with the patient’s clinical phenotype and, consequently, p.Asn323Lys in HSD3B2 was the best disease-causing candidate variant.

Sanger sequencing confirmed the variant at the homozygous state in the proband (II-3) and in his affected sister (II-4). Segregation analysis showed that each of the healthy parents were heterozygous for this variant, while the other siblings were homozygous for the wild-type allele (Figure 3C).

## 4. Discussion

Pathogenic variants in *HSD3B2* leading to 3βHSD2 deficiency cause adrenal insufficiency and sex hormone synthesis dysfunction responsible for external genital ambiguity. In the present study, we describe two newborn Moroccan twins presenting SW-CAH, correlated to decreased aldosterone and cortisol with high ACTH and renin. The biochemical diagnosis is further complicated in early infancy by the rapid age-related changes in the plasma concentrations of various steroids [12], and by the fact that advanced techniques for the complete characterization of adrenal steroids, such as liquid chromatography–tandem mass spectrometry (LC-MS/MS), are not always available in clinical centers. Thus, although the diagnosis of SW-CAH, such as 3βHSD deficiency, was suspected in both patients, molecular studies are necessary to identify the causative enzyme defect [12], and to make differential diagnoses with other forms of CAH, in particular 21-hydrohylase and 17-hydroxylase deficiency, that require different treatment and follow up options.

Analysis through array CGH-SNPs detected a 7 Mb duplication on chromosome 10q22.3-q23.2 and two regions of homozygosity (ROH), respectively, of 8.36 Mb on 1p13.2-p11.2 and of 7.27 Mb on 14q31.1-q32.11 (Figure 2). The aCGH was performed as first tier test because it is not uncommon to find CNVs correlated to DSD as revealed by diagnostic rates of 22– 30% when large DSD cohorts are screened using aCGH [13]. At the same time, the SNP genotyping performed through the same array platform represents in our experience, an efficient approach to disclose LCSHs [14].

By comparing genes within these two LCSHs with the OMIM database and literature, the best candidate to explain the patient’s phenotype was *HSD3B2* causing the 3βHSD2D recessive disorder. In a previous study, UPD of the entire chromosome 1 disclosed a mutation in *HSD3B2* in a 7-day-old male infant with SW-adrenal crisis, as the molecular basis for CAH [15]. In the present case, the homozygosity is only partial and confined to an interstitial region of 8.36 Mb that might represent an identity-by-descent genomic segment. Therefore, it is possible that the parents are distantly related, also considering the presence of another LCSH on 14q31.1-q32.11 and the fact that they are both originally from a small county in Morocco.

Currently, more than 50 mutations have been found in *HSD3B2* [6], among which missense variants are the most common type that might be associated with some residual enzymatic activity and non-SW phenotype. Both the affected twins described here, with a SW phenotype, are carriers of a missense variant (N323K) located in a highly conserved region through different mammalian species that may impair steroid binding, nicotinamide binding, or enzyme stability. The same variant was previously reported as a cause of a severe salt-wasting form of congenital adrenal hyperplasia but normal female genitalia, in two sisters born from consanguineous Moroccan parents [16], and in another Moroccan family with a male with penoscrotal hypospadias, and a female with no DSD who were both suffering from a salt-wasting crisis at the 15th day of life [17]. Furthermore, a missense pathogenic variant c.967A > G (p.N323D) that affected the same amino acid residue was previously described as causative of a classical salt-losing 3βHSD2 deficiency. The functional in vitro study showed that the p.N323D mutant exhibited less than 5% of wild type enzyme activity [18]. From all these evidences, we can hypothesize that N323K might result in the synthesis of an enzyme with a reduced reactivity that leads to the pathology of the two affected newborns.

The affected twins also carry a duplication on 10q22.3-q23.2, likely determined by the low copy repeats (LCRs) LCR3 and LCR4, involved in chromosomal rearrangements mediated by non-allelic homologous recombination (NAHR). Deletions within the same genomic region (10q22.3-q23.3 deletions) are associated with intellectual disability, mild facial dysmorphisms, cardiac defects, macrocephaly, cerebellar anomalies, and congenital breast aplasia [19]. Only 13 patients with overlapping duplications on 10q22.3-q23.3 have been reported in literature with dysmorphisms and speech and/or motor delays with variable penetrance [20]. Among the genes included in the duplicated region, three might be clinically relevant as they are involved in mendelian disorders, namely *BMPR1A*, *LDB3* and *NRG3*. Mono allelic pathogenic variants in *BMPR1A* are causative of juvenile polyposis syndrome (JPS) and congenital heart disease (CHD) [21]. Mutations in the *LDB3* gene that encodes a PDZ-LIM domain binding factor that plays an important role in maintaining the structural integrity of the striated muscle Z-disc, have been identified in some cardiomyopathies in multiple species, such as dilated cardiomyopathy (DCM), hypertrophic cardiomyopathy (HCM), and LV non-compaction (LVNC) [22]. Finally, *NRG3* has been implicated in early mammary gland development, and it has also been associated with schizophrenia risk [23]. Moreover, the here described duplication is included within the chromosomal region responsible for the partial trisomy 10q syndrome, a cytogenetically diagnosable condition characterized by a severe phenotype with developmental delay, craniofacial abnormalities, talipes, microcephaly and congenital heart disease [24]. Considering the phenotype of our patients that currently do not manifest any evident heart defects, dysmorphisms, or other congenital anomalies, and the fact that the duplication is inherited from the unaffected mother and is also present in a healthy brother, the significance of this CNV and his contribution to the clinical phenotype remains uncertain. However, duplications in this region have been reported with reduced penetrance and the identification of carriers was of particular relevance, as they will be followed-up in the future and evaluated for the correlated clinical feature, in particular, heart defects.

At present, the identification of the likely pathogenic variant in *HSD3B2* provided unequivocal indications about medical and surgical management and follow up, risk of family recurrence, and need for monitoring adrenal and gonadal function [25], including the risk of developing testicular adrenal rest tumors (TARTs) or, rarely, ovarian and extra-gonadal adrenal rest tumors [26]. Furthermore, prompt diagnosis allowed the prevention of a serious neonatal adrenal crisis, especially in the newborn girl with CAH and normal external genitalia, which could have a delayed diagnosis and escape the newborn 17OH progesterone screening due to slight 17OHP increase, especially without adrenal crisis in the first two weeks of life.

## 5. Conclusions

The combination of two large scale approaches (aCGH and Clincal Exome) allowed a precise and complete diagnosis immediately after birth by detecting the co-presence of a pathogenic variant and of a CNV, whose consequence is not evident at present, but might influence and worsen the clinical phenotype during growth.

In general, the joint detection of CNVs and SNVs (single nucleotide variants) should be performed in newborns with complex phenotypes, as it might disclose the co-presence of multiple genetic factors. Currently, such patients are diagnosed through a step by step approach, including aCGH [27], targeted gene panels [28], exome sequencing, and in some accurately selected cases Whole Genome Sequencing (WGS), which is still expensive and requires specialized bioinformatic skills. In the future, with the improvement of bioinformatic tools, training of specialized personnel, and cost reductions, the introduction of routine diagnostics of WGS, detecting any type of genetic alteration, will greatly improve the management of these patients.

## Figures and Tables

**Figure 1 genes-13-02190-f001:**
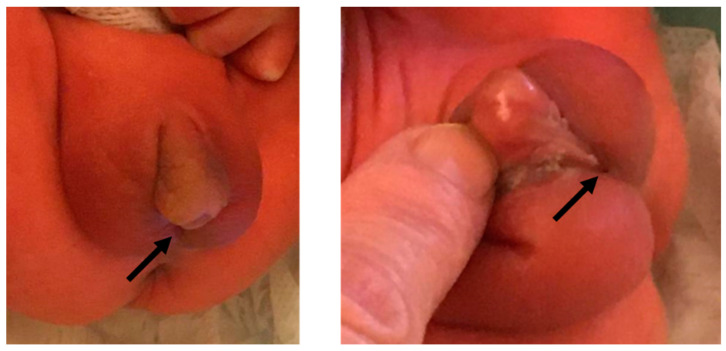
External genitalia of II-3 newborn at birth. Perineal hypospadias (black arrow) with small penis, hyper pigmented and fused scrotal folds containing palpable oval formations.

**Figure 2 genes-13-02190-f002:**
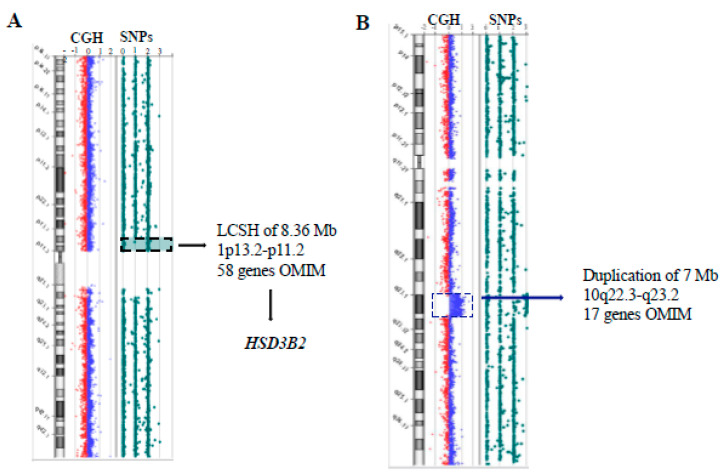
Array-CGH-SNP results. (**A**) The LCSH on chromosome 1p13.2-p11.2 is indicated by a black dashed rectangle. This region of 8.36 Mb containing 58 OMIM genes among which *HSD3B2*. (**B**) Duplication of 7 Mb,10q22.3-q23.2 (81,641,918_88,717,407)x3 which contains 17 OMIM genes.

**Figure 3 genes-13-02190-f003:**
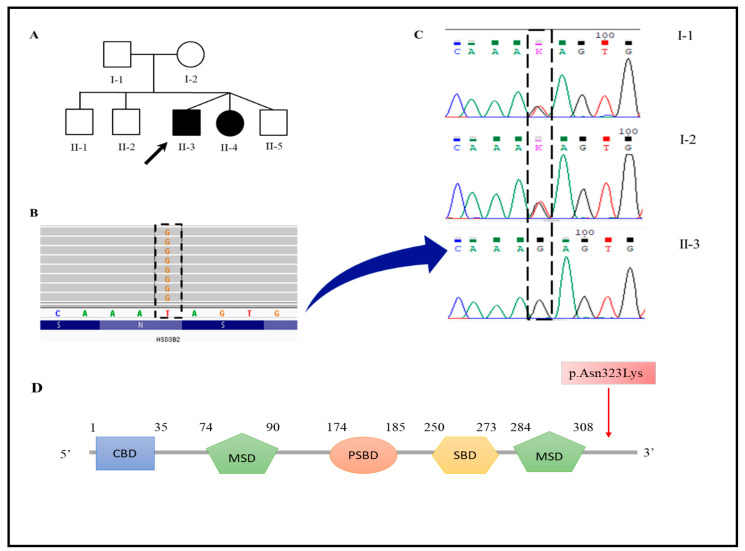
Pedigree, sequencing results and schematic presentation of *HSD3B2*. (**A**) Pedigree of the family, the black arrow indicates the proband investigated through CES and array CGH. (**B**) Representative image of reads alignment showing the missense variant in the *HSD3B2* gene (NM_000198.3:c.969T > G; p.Asn323Lys). (**C**) The variant c.969T > G was confirmed by Sanger sequencing. Segregation analysis revealed that the parents were heterozygous for the same variant. (**D**) Schematic diagram of the functional domains of *HSD3B2*, including cofactor binding domain (CBD), membrane-spanning domain (MSD), putative substrate-binding domain (PSBD) and substrate-binding domains (SBD). The red box shows the likely pathogenic variant detected in this study and located in the C-terminus.

**Table 1 genes-13-02190-t001:** Clinical, instrumental and laboratory findings of newborn triplets investigated for salt wasting congenital adrenal hyperplasia (SW-CAH).

	First Newborn	Second Newborn	Third Newborn
External genitalia	Hypovirilized male	Normal female	Normal male
Clinical appearance	Failure to thrive, eating difficulties and vomiting	Failure to thrive, eating difficulties and vomiting	Normal post-natal growth
Abdomen US scan:			
female internal genitalia	Absent	Present	Absent
external genitalia	Ovular structures	Normal female	Normal male
adrenal	Normal	Normal	Normal
Laboratory investigation:			
ACTH (pg/mL)	1115	580	32.7
Cortisol (mcg/dL)	2.9	4.1	8.4
17OHP serum (ng/mL)	>20	>20	1.39
17OHP whole blood (ng/mL)	>110	64	-
DHEAS (ng/mL)	>1000	958	-
D4androstenedione (ng/mL)	15.80	>10	-
Testosterone (ng/mL)	6.05	6.29	-
17beta-estradiol (pg/mL)	74.7	51.5	-
Renin (μIU/mL)	>500	756	147.1
Aldosterone (ng/dL)	54.7	14.6	330
Na+ (mEq/L)	133	129	-
K+ (mEq/L)	5.1	5.7	-
AMH (ng/mL)	>20	-	-

## Data Availability

Not applicable.

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
