# Peer review of "Co-Occurrence of a Pathogenic HSD3B2 Variant and a Duplication on 10q22.3-q23.2 Detected in Newborn Twins with Salt-Wasting Congenital Adrenal Hyperplasia"

_genes, 2022, doi:10.3390/genes13122190_

Round 1

Reviewer 1 Report

In this paper, the authors describe the case of two twins with a disorder of sex development harboring a germline HSD3B2 variant associated with a 7 Mb duplication on chromosome 10. The p.N323K HSD3B2 variant is most probably responsible for the disease as this variant has previously been reported in patients with congenital adrenal hyperplasia. Moreover, this variant is at homozygous state in both the proband and his affected sister whereas the parents are heterozygous and the non-affected siblings are homozygous wild-type. This case report confirms the importance of this HSD3B2 genetic variant in the development of congenital adrenal hyperplasia.

Minor comments:

-     Please remove the lines 31 to 38 that describe the information that should be included in the introduction.

-      The gene should be written in italics

-      The authors should refer to figure 2 in the text like for instance while they discuss the sanger results. However, based on the chronology of the figures cited in the text, figure 3 will become figure 2 and figure 2, figure 3.

Author Response

Dear Reviewer

Thanks for your suggestions. We changed the manuscripts according to your comments as follows

  • Please remove the lines 31 to 38 that describe the information that should be included in the introduction.

Yes, we apologize for this, it was an error.

  • The gene should be written in italics

Thanks, it has been done for all the genes cited in the manuscript (HSD3B2 and the others in the discussion) when the term refers to the gene and it is not followed by gene (e.g HSD3B2 gene)

3) The authors should refer to figure 2 in the text like for instance while they discuss the sanger results. However, based on the chronology of the figures cited in the text, figure 3 will become figure 2 and figure 2, figure 3.

-The numbers of the figures have been changed and corrected in the text

Reviewer 2 Report

This is a well written case report and I have only a few minor issues that should be addressed.

Page 4, line 151: throw should be through

Page 5, line 165: check sentence structure

Page 5, line 185: should be homozygosity

Page 5, line 197: I am not sure that I agree with the ACMG classification. The current ClinGen guidelines should be consulted and used. PM2 should be downgraded to supporting, PP2 for rare missense does not apply to HSD3B2 according to gnomAD (similar observed versus expected for missense), and PP1 at the level of very strong needs to be supported/explained. I am not sure the current family gets you to very strong (considering PMID: 27236918).  If you are using prior published data this should be provided. You could consider using PS4 at a reduced strength or PP4, it is also possible that PM3 could be utilized especially since you have literature reports of four other individuals reported with this variant (you need to be careful not to double count but I would consider using 1-2 of these criteria). I would typically use REVEL for PP3 but that does not work for this variant (0.3919).

Page 8, line 260: are missense variants “generally associated with some residual enzymatic activity”? I read PMID: 10599696 which presented a number of missense variants without residual enzymatic activity.

Page 8, line 272: I am not sure it can be assumed that N323K results in the synthesis of an unstable protein with probably loss of enzymatic activity based on N323D. This should not be said as strongly as it is stated. The same is true for line 277 suggesting that missense variants in this region are more severe, if this is true more evidence needs to be provided.

Page 9, line 316: It is not clear that aCGH was needed to diagnose these two individuals. Exome sequencing would have defined the regions of homozygosity which lead to the identification of the likely causative variant. The duplication is not clearly related to the phenotype and may not add anything to the treatment of these patients.

Author Response

Dear Reviewer

Thanks for your suggestions. We changed the manuscripts according to your comments as follows:

1)Page 4, line 151: throw should be through

Thanks, the entire sentence has been changed

2)Page 5, line 165: check sentence structure

We changed the sentence structure

3)Page 5, line 185: should be homozygosity

Thanks it was our mistake

4)Page 5, line 197: I am not sure that I agree with the ACMG classification. The current ClinGen guidelines should be consulted and used. PM2 should be downgraded to supporting, PP2 for rare missense does not apply to HSD3B2 according to gnomAD (similar observed versus expected for missense), and PP1 at the level of very strong needs to be supported/explained. I am not sure the current family gets you to very strong (considering PMID: 27236918).  If you are using prior published data this should be provided. You could consider using PS4 at a reduced strength or PP4, it is also possible that PM3 could be utilized especially since you have literature reports of four other individuals reported with this variant (you need to be careful not to double count but I would consider using 1-2 of these criteria). I would typically use REVEL for PP3 but that does not work for this variant (0.3919).

5)Thanks, the problem is that the classification changes over the time and it should be revised, so thanks for having suggested this point. The variant HSD3B2 c.969T>G is not present in ClinVar. We reclassified it with the ACMG criteria.  The variant is thus now classified as Likely Pathogenic  and  meets the following criteria:

-PP1: supporting, it co-segregates in two affected members of the family and it is located in a gene already known for being involved in this disorder.

-PM2: supporting, absent in the controls  

-PM3: moderate, it is present also in trans on the other allele, the affected individuals have two alleles carrying the mutation that were inherited by the two healthy parents

PM1:moderate, it is located in a well-established enzyme domain as demonstarated by Guran et al (ref 18) who reported a different variant at the same aminoacidic position  that reduced the enzyme activity to less than 5% of the wild-type.

PM5: moderate, the same aminoacid was altered with a different change (p.N323D)  in a patient described by Guran et al (ref 18) and this mutant showed less than 5% activity with respect to the wild-type

6)Page 8, line 260: are missense variants “generally associated with some residual enzymatic activity”? I read PMID: 10599696 which presented a number of missense variants without residual enzymatic activity.

Yes, we changed the sentence as follows: “….that might be  associated with some residual enzymatic activity and non-SW phenotype….”

7)Page 8, line 272: I am not sure it can be assumed that N323K results in the synthesis of an unstable protein with probably loss of enzymatic activity based on N323D. This should not be said as strongly as it is stated. The same is true for line 277 suggesting that missense variants in this region are more severe, if this is true more evidence needs to be provided.

8)Thanks for this comment. We change the sentences in the discussion regarding this point and said the same concept  less strongly and removed the sentence at line 277.  

Page 9, line 316: It is not clear that aCGH was needed to diagnose these two individuals. Exome sequencing would have defined the regions of homozygosity which lead to the identification of the likely causative variant. The duplication is not clearly related to the phenotype and may not add anything to the treatment of these patients.

9)Yes the duplication dected throught CGH array in a VUS, as we stated in the text and it has no an apparent influence on the phenotype. However, it might become relevant during the growth. The following sentences of the discussion explain this concept:

“……..The aCGH  was performed as  first tier test because it is not uncommon to find CNVs correlated to DSD as revealed by diagnostic rates of 22– 30% when large DSD cohorts were screened using aCGH……..”

“……..However duplications in this region  have been reported with reduced penetrance and the identification of carriers was of particular relevance as  they will  be followed-up  in the future and evaluated for the correlated clinical feature, in particular hearth defects……...”